# Diagnostic Errors in Japanese Community Hospitals and Related Factors: A Retrospective Cohort Study

**DOI:** 10.3390/healthcare11111539

**Published:** 2023-05-25

**Authors:** Taichi Fujimori, Ryuichi Ohta, Chiaki Sano

**Affiliations:** 1Faculty of Medicine, Shimane University, 89-1 Enya-cho, Izumo 693-8501, Japan; muchisimas.99@gmail.com; 2Oda Municipal Hospital, 1428-3 Yoshinaga, Oda-cho, Oda 694-0063, Japan; 3Community Care, Unnan City Hospital, 699-1221 96-1 Iida, Daito-cho, Unnan 699-1221, Japan; 4Department of Community Medicine Management, Faculty of Medicine, Shimane University, 89-1 Enya cho, Izumo 693-8501, Japan; sanochi@med.shimane-u.ac.jp

**Keywords:** diagnostic error, regional hospital, emergency medical care, cognitive bias, sex bias

## Abstract

Diagnostic error has recently become a crucial clinical problem and an area of intense research. However, the reality of diagnostic errors in regional hospitals remains unknown. This study aimed to clarify the reality of diagnostic errors in regional hospitals in Japan. A 10-month retrospective cohort study was conducted from January to October 2021 at the emergency room of Oda Municipal Hospital in central Shimane Prefecture, Japan. Participants were divided into groups with or without diagnostic errors, and independent variables of patient, physician, and environmental factors were analyzed using Fisher’s exact test, univariate (Student′s *t*-test and Welch’s *t*-test), and logistic regression analyses. Diagnostic errors accounted for 13.1% of all eligible cases. Remarkably, the proportion of patients treated without oxygen support and the proportion of male patients were significantly higher in the group with diagnostic errors. Sex bias was present. Additionally, cognitive bias, a major factor in diagnostic errors, may have occurred in patients who did not require oxygen support. Numerous factors contribute to diagnostic errors; however, it is important to understand the trends in the setting of each healthcare facility and plan and implement individualized countermeasures.

## 1. Introduction

Diagnostic errors are crucial clinical problems that affect patient prognosis. All patients should be diagnosed accurately and treated as soon as possible; however, diagnostic errors complicate this process. According to a study reported in the US, deaths from medical errors rank third after heart disease and malignant tumors, and diagnostic errors significantly affect patient prognosis [1]. In particular, diagnostic errors occur in 10–15% of cases [2] and account for many medical errors. While there is no universal definition of diagnostic error [3], one that is generally well-established is that by the National Academy of Medicine, “failure to (a) establish an accurate and timely explanation of the patient’s health problem(s) or (b) communicate that explanation to the patient;” the World Health Organization also adopts this definition.

There are many classifications of diagnostic errors that vary according to a report. For example, they may be categorized as no-fault, system-related, cognitive, and others [2]. Several studies have reported common diagnostic errors. The frequency of diagnostic errors varies depending on the type of patient and criteria selected; however, in one American study, when calculated, 5.08% of adult outpatients (i.e., 12 million patients annually) experience a diagnostic error [3]. Graber et al. also reported a 10–20% rate of diagnostic errors in autopsy cases that could have changed the treatment plan [4]. However, diagnostic errors are harmful and costly. In one study that aggregated malpractice claims, diagnostic errors accounted for the largest proportion of all lawsuits (28.6%), impacted patient outcomes, and resulted in the highest total claims (35.2%) [5]. In a study on medical malpractice lawsuits due to diagnostic errors in the United States, diagnostic errors were more likely to result in death than other factors (40.9% vs. 23.9%). Diagnostic errors are also the most common cause of lawsuits related to death or disability [6].

As noted above, there are many reports that diagnostic errors are common, costly, and harmful, and they are now one of the major problems that should be addressed to improve the quality of medical practice. When a diagnostic error occurs, the person involved tends to be held accountable for the event [7]; however, only 4% and 7% of cases are due to a lack of knowledge and diagnostic skills, respectively [2]. However, as a countermeasure, it is inadequate to merely study these factors or call on the person to reflect on his past conduct. Some of the classifications of the causes of diagnostic errors are “situational factors” (physician’s stress, time zone, work style, equipment, and manpower), “information collection factors” (such as information gathering process and its interpretation obtained from too little or too much medical history collection, laboratory test, and physical examination), and “cognitive bias” (such as intuition). Hospital size, location, and patient-to-medical ratio are also crucial situational factors.

Because the causes of diagnostic errors are diverse, various preventive methods have been reported. These include teamwork, such as multidisciplinary and patient collaboration [8]; improvement of individual performance; health information technology, such as electronic medical records [9]; the role of organizations; and education [10]. Various studies have been conducted on diagnostic errors, but few investigations have been conducted in local hospitals. In local hospitals, even if a diagnostic error occurs, it is not tracked and analyzed, and because there is no chance for a retrospective evaluation, no preventive or improvement measures are taken. Some staff members may not even be aware that a diagnostic error has occurred if it does not cause critical outcomes. Therefore, the actual state of diagnostic errors should be clarified and improved for better diagnosis and treatment in local hospitals.

We can create a foothold for establishing diagnostic error prevention methods by investigating diagnostic errors in local hospitals, determining their characteristics, and considering their causes and preventive measures. We considered the characteristics of diagnostic errors in medium-sized hospitals and discussed the types of effective preventive measures.

## 2. Materials and Methods

### 2.1. Method

This was a retrospective cohort study. The survey period was 10 months from January to October 2021. This study was approved by the ethics committee.

### 2.2. Setting

The patients who visited the emergency room (ER) of Oda Municipal Hospital (229 beds (139 beds for acute phase, 45 beds for convalescent rehabilitation, 45 beds for integrated community care system); 20 departments) in Oda City, located in the central Shimane Prefecture, Japan, were included in the study. An ER in Japan is utilized in various ways and often refers to something different from the international concept. In the ER of Oda Municipal Hospital, all emergency patients are first examined by the doctor in charge of the ER on that day, regardless of severity or type of injury. There is no emergency physician, and when a patient is hospitalized, the doctor who first examined the patient at the ER becomes the attending doctor. The population of Oda City is approximately 34,000. It is a medium-sized medical institution that provides medical care to approximately 53,000 people, including a population of just under 19,000 living in the surrounding area. Approximately 38% of the population is aged around 65 years. The total number of patients who visited the ER from April 2020 to March 2021 was 5654 (monthly average 471.2, daily average 15.5). Among these, 2616 were male and 3038 were female. There were 1707 inpatients and 3947 outpatients. Of these, 184 were transferred to another hospital.

### 2.3. Participants

Patients who visited the ER were under the care of a physician and were either hospitalized or transported to a different hospital. Although no established definition for diagnostic error exists, the criteria for judging a diagnostic error in this survey were as follows: cases admitted to Oda City Hospital, cases where the diagnosis was modified between admission and discharge, or cases in which no diagnosis was made. Cases that were transported to another hospital were considered cases wherein the diagnosis provided at the Oda City Hospital differed from that provided at the destination. However, to solely target as many medical cases as possible, obstetrics cases; COVID-19 cases (including those who wished only for tests for COVID-19); surgical cases including trauma; minor dermatological cases such as urticaria, stings by bees, and bites by a poisonous snake; visits for injections only; emergency visits due to side effects of vaccination; and children under 6 years of age were excluded from the survey. We also excluded cases in which the diagnosis was clear during consultation, such as terminal cancer hospitalization (Figure 1).

### 2.4. Data Collection

#### 2.4.1. Primary Outcome

The primary outcome was the occurrence of diagnostic errors. In the group with diagnostic errors (DE group), a “delayed diagnosis” was set when the diagnosis or explanation to the patient/family was not appropriately or promptly made. If the correct diagnosis was overlooked or excluded, it was considered a “missed diagnosis”, and when an inaccurate diagnosis was provided, it was marked as a “wrong diagnosis”.

#### 2.4.2. Independent Variable

Various causes of these diagnostic errors have been reported. Among them, cognitive bias accounts for approximately 90% of diagnostic error factors in ERs. Therefore, in this survey, we investigated patient, physician, and environmental factors linked to cognitive bias [11,12,13,14].

##### Patient Factors

Age, sex, vital signs (level of consciousness, body temperature, blood pressure, and heart rate), initial diagnosis (diagnosis in the ER), last diagnosis (diagnosis based on post-hospital care), with or without oxygen support, number of medications, body mass index (BMI), age-adjusted Charlson comorbidity index (CCI), and outcome were measured. There is no strict definition as to what number of medications is considered polypharmacy. However, according to a report investigating the relationship between the number of medications taken and adverse drug events in elderly hospitalized patients, the frequency of adverse drug events increased, particularly with six or more drugs [15]. Considering these results and actual prescribing to the elderly, the polypharmacy cutoff for this study was set at ≥6 medications.

##### Physician Factors

Whether the attending physician was a junior resident was noted. In addition to junior residents, the number of specialists, such as gastroenterologists, cardiologists, dermatologists, general practitioners, orthopedic surgeons, pediatricians, endocrinologists, anesthesiologists, surgeons, radiologists, urologists, physicians, neurologists, respiratory physicians, and neurosurgeons, who attended to the case was noted as well.

##### Environmental Factors

We tabulated the time zones in which the patients visited. The day shift was from 9:00 a.m. to 6:00 p.m., and the night shift was from 6:00 p.m. to 9:00 a.m. the next day.

#### 2.4.3. Analysis

Univariate and logistic regression analyses were performed. In univariate analysis, continuous variables were analyzed using Student’s *t*-test, Welch’s *t*-test, and nominal variables using Fisher’s exact test. Welch’s *t*-test was used for age (all cases and females). In univariate regression analysis, age, blood pressure, heart rate, and BMI were analyzed as continuous variables. Meanwhile, in the logistic regression analysis, similar to any other item, these items were converted to nominal variables and analyzed in consideration of suitability. Numerical variables were dichotomized as follows: age, ≥75 years and <75 years; body temperature, ≥37.2 °C and <37.2 °C [16]; number of medications, ≥6 and <6; age-adjusted CCI [17], ≥5 and <5; time zones, from 9:00 a.m. to 6:00 p.m. and from 6:00 p.m. to 9:00 a.m. the next day; systolic blood pressure, ≥150 mmHg and <150 mmHg; diastolic blood pressure, ≥90 mmHg and <90 mmHg; heart rate, X ≤ 60 or X ≥ 100 and 60 < X < 100 [18]; and BMI, X ≤ 18.5 or X ≥ 25 and 18.5 < X < 25 (Adapted from World Health Organization 2023).

#### 2.4.4. Ethics

This study was conducted in accordance with the principles of the Declaration of Helsinki. The hospital was assured of the anonymity and confidentiality of the patient information used in this study. Information related to this study was posted on the hospital website without disclosing any patient details. The contact information of the hospital and the research representative was also listed on the website to ensure that any questions regarding this study were addressed. All participants were informed of the purpose of this study; they provided informed consent and were provided with the opportunity to decline participation at any time. The study was approved by the ethics committee of Oda City Hospital (approval date: 1 February 2022).

## 3. Results

### 3.1. Background Results

#### 3.1.1. Details of Diagnostic Errors

We reviewed 4803 medical records in all. The total number of eligible patients was 924. Furthermore, 121 cases met the definition of diagnostic error, accounting for 13.1% of the eligible cases. We categorized the diagnostic error cases as delayed, wrong, and missed, which comprised 30.6% (37/122), 37.2% (45/122), and 33.1% (40/122) of the patients, respectively. In the DE group, “no diagnosis” was most common for both the initial and final diagnoses (30.5% and 23.1%, respectively). We aggregated the independent variables (patient and environmental factors) for each group.

#### 3.1.2. Univariate Analysis Results for Independent Variables

As a result of univariate analysis for independent variables (Table 1), the average age was 74.0 vs. 79.3 years old (*p* = 0.001), the proportion of female patients was 43.8 vs. 52.1% (*p* = 0.098), the proportions of cases with NOT alert was 19.0 vs. 31.4% (*p* = 0.005), the proportions of cases with fever above 37.2 °C was 35.6 vs. 36.1% (*p* = 1), the average of systolic blood pressure was 141.7 vs. 135.3 mmHg (*p* = 0.037), the average of diastolic blood pressure was 79.7 vs. 76.5 mmHg (*p* = 0.070), the average of heart rate was 84.0 vs. 87.6 mmHg (*p* = 0.102), the proportion of cases with oxygen support was 8.0 vs. 22.2% (*p* = 0.0002), the average of BMI was 22.6 vs. 21.4%, the proportion of patients with age-adjusted CCI ≥ 5 was 71.1 vs. 77.4% (*p* = 0.135), the proportion of patients taking six or more concomitant medicines was 52.5 vs. 58.8% (*p* = 0.238), the proportion of cases whose clinical outcome was death was 6.7 vs. 11.5% (*p* = 0.154), and the proportion of cases who visited during the night shift was 33.1 vs. 34.1% (*p* = 0.838) in the DE and NO DE groups, respectively. Univariate analysis revealed significant differences in five parameters: age, level of consciousness, systolic blood pressure, BMI, and oxygen support.

#### 3.1.3. Independent Variables/Physician Factor and Environmental Factor

Regarding independent variables/physician factors, we aggregated the frequency of involvement of physicians in the initial diagnosis: 21.5 vs. 24.0% for general practitioners and 22.3 vs. 30.6% for residents in the DE and NO DE groups, respectively. However, no significant difference was observed. Gastroenterologists and cardiologists were more likely to be involved in cases of diagnostic errors, whereas residents and surgeons were less likely to be involved. As for independent variables and environmental factors, we investigated the time zones in which the patients visited the ER. There was no particular difference in the occurrence of diagnostic errors between the day and night shifts.

### 3.2. Logistic Analysis Results

Logistic analysis with 13 independent variables showed that sex and oxygen intake differed significantly between the two groups (*p* = 0.029 and 0.010, respectively). Age (*p* = 0.080), BMI (*p* = 0.735), consciousness level (*p* = 0.454), fever (*p* = 0.890), systolic BP (*p* = 0.116), diastolic BP (*p* = 0.924), HR (*p* = 0.971), number of medications (*p* = 0.265), outcome (*p* = 0.591), time zone (*p* = 0.759), and age-CCI (*p* = 0.077) were not significantly different between the two groups (Table 2).

## 4. Discussion

### 4.1. Summary

We investigated the frequency of diagnostic errors in regional hospitals in Japan. In Oda City Hospital, 13.1% of all eligible cases had diagnostic errors. Further, we investigated the differences in background factors with and without diagnostic errors. Interestingly, logistic regression analysis showed that the proportion of patients treated without oxygen support was significantly higher and that of female patients was significantly lower in the group with diagnostic errors than in the group without. This is discussed in detail below.

### 4.2. Reality of Diagnostic Errors in Regional Hospitals in Japan

Diagnostic errors accounted for 13.1% of the cases. Although the frequency of diagnostic errors varies by study, previous studies have reported it to be around 10–15% [2] and 10–20% [19] of cases; the results of these studies are almost consistent with our results. However, when limited to the ER, diagnostic errors occurred in 0.6–12% of the cases [20,21,22], which is slightly lower than that in our report. Thus, the frequency of diagnostic errors in the ER at Oda City Hospital is average; however, given the ER setting, it is considered slightly high. The most probable reason is the lack of a feedback system for diagnostic errors. Therefore, no measures such as extraction, analysis, and reporting of diagnostic errors; dissemination of countermeasures; enlightenment; education; or cooperation in the event of errors were captured. It is considered adequate to disseminate the results of this study to improve diagnostic accuracy. Numerous facilities may possibly provide medical care in similar environments. Therefore, disseminating the results to medical personnel at these facilities would effectively reduce the frequency of diagnostic errors both domestically and internationally. It is also important to take this opportunity to establish countermeasures according to the hospital environment [23]. However, there are no acute care physicians in the ER of Oda Municipal Hospital. This situation, often seen in regional hospitals in Japan, means that non-acute care physicians such as radiologists and dermatologists take turns in charge of the ER. Although there are many studies on ERs around the world, it is important to note that the ER in this study may differ from others, which may be a contributing factor to the slightly high diagnostic error rate.

### 4.3. Sex

Background factors, including sex, lifestyle, and social status, are often considered when diagnosing patients. Sex bias has often been the subject of attention in various clinical situations. For example, it has been pointed out that because much traditional medical research has been conducted on men, many issues related to women’s health have been overlooked.

Conversely, in diseases such as osteoporosis and multiple sclerosis, where research is biased toward actively recruiting only female participants, there are also claims that the quality of medical care for men is not maintained [24]. Logistic regression analysis revealed that the proportion of female patients was significantly lower in the group with diagnostic errors. Although the results of this study alone cannot prove the cause of the significant difference in the data, at least we can conclude that there is a sex bias in the occurrence of diagnostic errors. The fact that women are less susceptible to diagnostic errors can be expected to contribute to the survival rate, prognosis, and quality of life, making this an important future research question.

### 4.4. Oxygen Support

Logistic regression analysis showed that the proportion of patients treated without oxygen support was significantly higher in the group with diagnostic errors than in the group without diagnostic errors. Additionally, although there was no significant difference in the logistic regression analysis for age and level of consciousness, a significant difference was observed in the univariate analysis for each factor. Hence, as a first impression, when the physician judges the illness to be mild, there is a possibility of some cognitive bias in the mind of the attending medical staff (e.g., they tend to become more relaxed and turn their attention or devote more time to patients with more severe presentations). A previous study reported that an initial diagnosis error was common in mild diseases [25]. A factor analysis study of diagnostic errors in ERs revealed that cognitive bias accounted for approximately 90%, atypical extraordinary medical history for approximately 50%, information gathering factors for approximately 20–40%, systemic factors for approximately 20%, emotional factors for approximately 15%, the context of care for approximately 15%, and communication factors for approximately 10% of the cases [21,22,23]. It is evident from this study that cognitive biases are frequently involved in diagnostic errors in ERs. Cognitive biases may occur during information integration, and “early closure” and “misjudging the salience of a finding” have been reported to frequently occur among cognitive biases [2,26]. In this study, it is assumed that many cases were regarded as mild and closed early. Additionally, other biases may also be involved, such as confirmation bias, which attempts to find corroborative evidence that supports a hypothesis rather than finding evidence that rejects the hypothesis, and heuristic bias, which cuts off the thinking process primarily based on empirical rules. The analysis of each case revealed new trends. It may be helpful to additionally consider the diagnostic errors that are likely to occur when vital signs are stable.

### 4.5. CCI

Logistic regression analysis showed no significant differences in the CCI. However, because age-adjusted CCI was used in this study, there is a possibility that the scores in the DE group were lower than those calculated, although it was a result of univariate regression analysis, considering that the participants in the DE group were young. Diagnosis error cases may have a relatively high CCI if the age factor is excluded. Thus, the participants in the DE group may have more comorbidities that contribute to death. The more complex the comorbidities, the more difficult the assessment, which may distort the evaluator’s judgment. Lack of knowledge and diagnostic skills are responsible for 4% and 7% of diagnostic errors, respectively [2]. CCI may relate the evaluator’s knowledge and skill level to the success or failure of the diagnosis. Further investigation and analysis are required in this area.

### 4.6. Limitations

First, the definition of diagnostic errors was a limitation. Numerous previous studies of diagnostic errors have investigated autopsies and litigation cases. However, because of this method, the number of cases that can be included in a single-center study becomes too small. Consequently, the investigator (first author) reviewed the medical records and identified the diagnostic error criteria. Moreover, subjective research is arbitrary and subject to a high degree of bias. Therefore, we set a simple definition for diagnostic errors in this study and selected them as objectively as possible. However, some cases would likely be considered with diagnostic errors if other definitions were used but were judged to be without diagnostic errors, and vice versa. Different definitions may have yielded different results. A revised, safer Dx instrument [27] was developed to objectively judge whether a diagnostic error has occurred. This tool requires somewhat complicated information collection and was not adopted in this study because of a lack of manpower. Therefore, a study using this index might have been more helpful. Second, there was a lack of relevant data. For example, if a patient was transferred to another hospital, determining whether a diagnostic error occurred was not possible in cases where the transport destination provided no information. Third, various factors are associated with diagnostic errors. However, we did not collect information on any of these factors. For example, environmental factors, such as the number of patients being treated at the same time, severity, and urgency, were not considered, and there were limitations to a retrospective study.

## 5. Conclusions

In this study, we collected and analyzed data on diagnostic errors in the ER of a regional hospital in Japan. Our hospital does not take any special measures, specifically for diagnostic errors. Given the ER setting, the frequency of diagnostic errors was high; therefore, countermeasures were necessary. Numerous factors cause diagnosis errors; therefore, it is important to individually plan and implement countermeasures. The dissemination of the results of this research at Oda City Hospital and the opportunity for discussion will improve the quality of medical care in the medical prefecture area. Additionally, it is conceivable that numerous facilities provide medical care in a similar environment, and the results of this study may guide medical staff and patients at those facilities.

## Figures and Tables

**Figure 1 healthcare-11-01539-f001:**
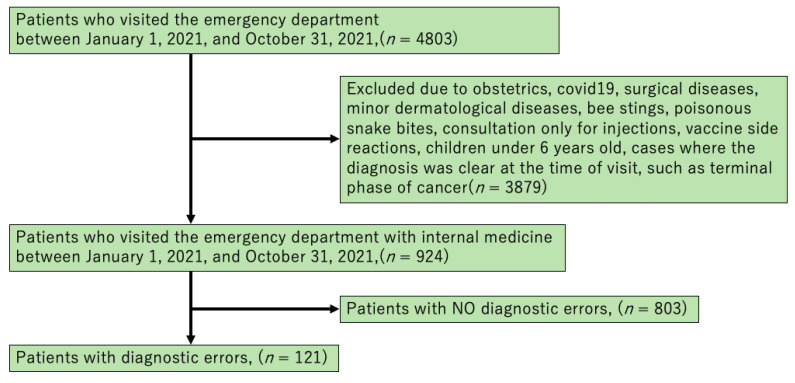
Flow chart of patient selection.

**Table 1 healthcare-11-01539-t001:** Demographic characteristics of the cases in the univariate analysis.

	Diagnostic Error				
Factor	+		−		
		*n* (Total = 121)		*n* (Total = 803)	*p*-Value
Age, mean (SD)	74.02 (18.95)	121	79.30 (16.03)	803	0.004 **
Age (male), mean (SD)	72.60 (18.12)		76.29 (16.35)		0.092
Age (female), mean (SD)	75.85 (20.00)		82.06 (15.25)		0.033 *
Female patients (%)	53 (43.8)		418 (52.1)		0.098
NOT alert (%)	23 (19.0)	121	252 (31.4)	803	0.005 **
Fever (%)	37 (35.6)	104	269 (36.1)	746	1
Oxygen (%)	9 (8.0)	112	168 (22.2)	758	0.0002 **
No. of medications ≥ 6 (%)	53 (52.5)	101	428 (58.8)	728	0.238
Dead patients (%)	8 (6.7)	120	90 (11.5)	784	0.154
Systolic BP, mean (SD)	141.7 (32.88)	117	135.3 (30.43)	778	0.037 *
Diastolic BP, mean (SD)	79.7 (18.10)	116	76.5 (17.87)	774	0.070
HR, mean (SD)	84.00 (21.09)	108	87.59 (21.30)	744	0.102
BMI, mean (SD)	22.64 (4.02)	113	21.36 (4.03)	771	0.002 **
Age-CCI ≥ 5 (%)	86 (71.1)	121	609 (77.4)	787	0.135
Visits on night shift (%)	40 (33.1)	121	274 (34.1)	803	0.838

Fisher’s exact test, Student’s *t*-test, and Welch’s *t*-test. The total number does not always equal the number of cases “*n*”, owing to missing data in some cases. All participants assessed as having a level of consciousness other than alert were counted as “not alert”. BP, blood pressure; HR, heart rate; BMI, body mass index; age-CCI, age-adjusted Charlson comorbidity index. * *p* < 0.05; ** *p* < 0.01.

**Table 2 healthcare-11-01539-t002:** Results of the logistic regression model.

Factor	OR	95% CI	*p*-Value
Age ≥ 75 (reference: age < 75)	2.020	0.92–4.14	0.080
Female patients (reference: male patients)	0.566	0.34–0.94	0.029 *
NOT alert (reference: alert)	0.800	0.45–1.43	0.454
Fever (reference: body temperature < 37.2 °C)	1.040	0.61–1.78	0.890
Oxygen (reference: without oxygen support)	0.309	0.13–0.76	0.010 **
No. of medications ≥ 6 (reference: no. of medications < 6)	0.693	0.40–1.19	0.185
Dead patients (reference: alive patients)	1.300	0.50–3.33	0.591
Systolic BP (reference: systolic BP < 150 mmHg)	1.630	0.89–3.01	0.116
Diastolic BP (reference: diastolic BP < 90 mmHg)	1.030	0.52–2.05	0.924
HR: X ≤ 60 or X ≥ 100 (reference: 60 < X < 100)	0.990	0.58–1.69	0.971
BMI: X ≤ 18.5 or X ≥ 25 (reference: 18.5 < X < 25)	0.916	0.55–1.53	0.735
Age-CCI ≥ 5 (reference: age-CCI < 5)	0.482	0.21–1.08	0.077
Visits on night shift (reference: visits on day shift)	0.919	0.54–1.58	0.759

Fisher’s exact test. OR, odds ratio; CI, confidence interval; CCI, Charlson comorbidity index. * *p* < 0.05; ** *p* < 0.01.

## Data Availability

The datasets used and/or analyzed during this study may be obtained from the corresponding author upon reasonable request.

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
