# Peer review of "Diagnostic Errors in Japanese Community Hospitals and Related Factors: A Retrospective Cohort Study"

_healthcare, 2023, doi:10.3390/healthcare11111539_

Round 1
Reviewer 1 Report
This manuscript is a very interesting study and an important effort to improve clinical outcomes related to diagnostic errors.
Please clarify the statement of research aim in the Abstract and Introduction sections.
Clear research aims can provide clear research conclusions.
Please take the following points into consideration and revise them.
Unnecessary capitalization and abbreviations were used.
In addition, in the letters in the table, please unify the capitalization so that all letters begin with a capital letter.
Also, when using abbreviations, please provide an explanation of the abbreviation.
Line102
There are already several papers on the definition of diagnostic errors (please refer to the following). Please improve the text of the criteria for diagnostic errors used in this study.
https://www.ncbi.nlm.nih.gov/books/NBK338594/
https://journals.lww.com/journalpatientsafety/Fulltext/2022/12000/Defining_Diagnostic_Error__A_Scoping_Review_to.7.aspx
There are two identical expressions in the figure. Please delete "Figure. Flow chart of patient selection" above the figure, as it is redundant.
Line 144
The authors mentioned that "Univariate and logistic regression analyses were performed."
In fact, it is assumed that an independent t-test (Welch's t-test is appropriate here) was performed for the quantitative variables that were divided into two groups. Please refer to the following reference.
https://www.voxco.com/blog/welchs-t-test/
https://libguides.library.kent.edu/spss/independentttest
https://rips-irsp.com/articles/10.5334/irsp.82
How did you analyze the values expressed as percentages in Table 1?
Perhaps you performed a chi-square test, but please describe the exact statistical treatment method.
Also, how did you analyze the values indicated as percentages in Tables 1 and 2?
I assume you probably performed a chi-square test, but please describe exactly how you performed the statistical processing.
Also, please describe the statistical treatment method under Tables 1 and 2.
The authors state that "The total number does not always equal the number of cases "n," owing to missing 200 data in some cases. However, an exact "n" should be given for each case so that other researchers can analyze it.
Table 2.
I believe that the logistic regression analysis is interpreted differently.
Logistic regression analysis does not analyze significant differences.
Female patients (reference: Male patients) OR=0.566, 95% CI= 0.34-0.94, p=0.029
This value indicates that "the effect (association) for misdiagnosis is lower for female patients compared to male patients." This is also the case for oxygen intake.
On the other hand, Table 2 shows that "no variable was found to be associated with the risk of diagnostic errors."
Age, level of consciousness, systolic blood pressure, BMI, and oxygen support," which were found to be significantly different in the bivariate significance tests, were not applicable as risk factors in the logistic regression analysis. I believe that, for level of consciousness, systolic blood pressure, and BMI, these risk factors for diagnosis have been identified that you can assume that physicians are using this information for an accurate diagnosis.
Additionally, it is worth noting that the nighttime hours were not a risk factor for misdiagnosis. Can we not assume that your hospital has a good nighttime working system support (team care system)?
Please improve your statistical processing methods, your results, and further improve your discussion. This would make a good research report.
We would be happy to help. Please describe.
The authors state that "The total number does not always equal the number of cases "n," owing to missing 200 data in some cases. However, an exact "n" should be given for each case so that other researchers can analyze it.
Table 2.
I believe that the logistic regression analysis is interpreted differently. Logistic regression analysis does not analyze significant differences.
Female patients (reference: Male patients) OR=0.566, 95% CI= 0.34-0.94, p=0.029
This value indicates that "the effect (association) for misdiagnosis is lower for female patients compared to male patients." This is also the case for oxygen intake.
On the other hand, Table 2 shows that "no variable was found to be associated with the risk of diagnostic errors."
Age, level of consciousness, systolic blood pressure, BMI, and oxygen support," which were found to be significantly different in the bivariate significance tests, were not applicable as risk factors in the logistic regression analysis. I believe that, for blood pressure and BMI, those risk factors for diagnosis have been identified so that you could discuss that physicians are using this information for an accurate diagnosis.
Additionally, it is worth noting that the nighttime hours were not a risk factor for diagnostic errors. What do you think your hospital has a good nighttime working system support (interdisciplinary team care system)?
Please improve your statistical processing methods, your results, and further improve your discussion. This procedure would make a good research report.
I hope you will find it useful.
Reviewer 2 Report
Thank you for giving me the opportunity to review this manuscript. I have a few comments for the authors to consider
My primary concern is the methods section. it is unclear to me what it is the survey exactly? how was it administered ? and how is this study a retrospective one?
other comments:
- in the introduction section (line 63-73) please add references.
- please add a clear research question or objectives to the study.
- Why did the authors pick 6 medications to be a cut point ? it is very uncommon.
- please explain CCI to the readers.
- There are few terminologies that require revising such as time zone.
-the discussion section should be revised. Some subheadings need to be expanded on (e.g., CCI)
Round 2
Reviewer 1 Report
I am pleased that your manuscript has been improved.
However, I believe the following points need further improvement.
Line 12-17
You can revise the sentences as below:
This study aimed to clarify the reality of diagnostic errors in regional hospitals in Japan. A 10-month retrospective cohort study was conducted from January to October 2021 at the emergency room of Oda Municipal Hospital in central Shimane Prefecture, Japan. Participants were divided into groups with or without diagnostic errors, and independent variables of patient, physician, and environmental factors were analyzed using Fisher’s exact test, univariate (Welch’s t-test), and logistic regression analyses.
Line 142-143
You can revise the sentences as below:
Univariate and logistic regression analyses were performed. In univariate analysis, continuous variables were analyzed using Welch's t-test and nominal variables using Fisher's exact test.
Line 175
What is the meaning of the NOT alert?
Line 185
The authors mentioned that "Univariate analysis revealed significant differences."
What statistical analysis methods were used in the univariate analysis?
First group:
Sample size = 121
Mean = 74.0200
Standard deviation = 18.9500
Second group:
Sample size = 803
Mean = 79.3000
Standard deviation = 16.0300
Test of Equal Variance for Two Groups
F value = 1.39750
Degrees of freedom = (120, 802)
P-value = 0.010744 (two-tailed probability)
Normal t-test (when equal variances can be assumed)
t-value = 3.29353
Degrees of freedom = 922
P-value = 0.00103
When equal variances cannot be assumed (Welch's method)
t-value = 2.91194
Degrees of freedom = 147.01733
P-value = 0.00415 (exact value corresponding to fractional degrees of freedom)
Based on a two-tailed F test, σ1 (n=121) is considered as unequal to σ2 (n=803) (p-value is 0.0107). Since p-value < α, H0 is rejected.
The average of Group-1's population is considered to be not equal to the average of Group-2's population.
Thus, you have selected the Normal t-test (that is Student t-test), but you have to select the analysis result of Welch's method.
Line 196
There are still some unclear areas in the explanatory text for Table 1.
What is NOT alert?
Are there any data in Table 1 that show differences in the level of awareness?
There are other minor errors, so please check the comments in the pdf and correct them carefully.
Thank you so much.

Reviewer 2 Report
Thank you for addressing my comments.
Polypharmacy cutoffs often are often set at 5 medications. Only handful of studies that used 6 as a cutoff.
please review this article
https://bmcgeriatr.biomedcentral.com/articles/10.1186/s12877-017-0621-2
That being said, I highly recommend the authors to make it clear for the readers that they chose 6 medications as a cutoff and provide a reference.
